# Calibration and Test of Contact Parameters between Chopped Cotton Stalks Using Response Surface Methodology

**Bingcheng Zhang [1]** **, Xuegeng Chen [1], Rongqing Liang [1], Xinzhong Wang [2], Hewei Meng [1,\*] and Za Kan [1]**

[1] College of Mechanical and Electrical Engineering, Shihezi University, Shihezi 832000, China
[2] School of Agricultural Engineering, Jiangsu University, Zhenjiang 212013, China
\* Correspondence: mengbai4251982@sina.com; Tel.: +86-13369935035

**Abstract:** The accuracy of the material parameter settings directly affects the reliability of the results of the discrete element method simulation. It is necessary to calibrate the relevant parameters to obtain accurate discrete element simulation results when separating the cotton stalk particles from the residual film after crushing. The repose angle of the chopped cotton stalk particles was used as the response value to calibrate the contact parameters between particles. Physical tests measured the intrinsic particle and contact parameters between the cotton stalk particles and the contact material, which provided data for the simulation tests. According to the biological structure characteristics of cotton stalk, the discrete element method model of cotton stalk particles was constructed by bonding the elements of nonequal-diameter basic particles. Based on the response surface methodology, the stacking test of particles was simulated. The response model between the contact parameters and repose angle was established, and the effect law of the single-factor terms and interaction terms on the repose angle was analyzed. The optimal combination of contact parameters was obtained through the single-objective and multi-variable optimization methods. Finally, the contact parameter combination was verified by a simulation test of the repose angle. The results showed that the average relative error of the repose angle between the simulation test and the physical test was 1.04%, which verified the accuracy of the calibrated contact parameters and the reliability of the simulation test. These parameters provide a basis for the discrete element simulation study of cotton stalk motion in the separation process of cotton stalks and residual film and the subsequent gas–solid coupling simulation research.

**Keywords:** discrete element method; cotton stalk particles; contact parameters; calibration; response surface methodology

## 1. Introduction

Xinjiang, China's major cotton-producing area, commonly uses film mulching for cotton growing due to its geographical and environmental constraints [1–3]. Over the years, this growing method has increased cotton production and income in Xinjiang at the expense of serious residual plastic mulch pollution in farmlands [4,5]. For this reason, universities and research institutes have conducted research on the mechanized recycling of residual film, helping to alleviate such pollution to some extent. However, the recovered residual film mixed with cotton stalks makes reusing it more difficult [6,7]. Therefore, the separation of the residual film and impurities is necessary.

The cotton stalk is a major component of residual film mixtures. Separation from the residual film through conventional test methods makes it difficult to accurately analyze its force and movement. As computer technology advances, computer simulations are increasingly used in the design and optimization of agricultural machinery, greatly improving the efficiency of machinery development. The simulation technology based on the discrete element method (DEM) is widely used in developing and optimizing agricultural machinery [8,9]. The first step of a DEM-based numerical simulation is to establish a DEM

model and define its intrinsic parameters (such as density, Poisson's ratio, and elastic modulus, etc.) and contact parameters (such as the coefficient of restitution, coefficient of static friction, and coefficient of rolling friction, etc.) [10]. The first step of a numerical simulation in EDEM (DEM Solutions, United Kingdom) is to calibrate the parameters of materials; the DEM model's reliability depends on the contact model's accuracy and the parameters chosen for the particle properties. The intrinsic parameters of the materials can be obtained through physical experiments [11]. However, due to factors, such as the test methods, instruments, working conditions and constitutive law for particulate matter (particle shape, size, roughness), the contact parameters of the materials may be inaccurate, resulting in an inconsistency between the simulation results and actual test results [12]. In this case, it is necessary to define and calibrate the contact parameters of cotton stalks first when simulating the separation of the cotton stalks and residual film in EDEM so that the simulation results are consistent with the actual test results of the particle material in a specific area or under specific conditions.

DEM-based research on the physical characteristics and motion law of agricultural materials can provide a theoretical basis for designing agricultural machinery and equipment, which has become a trend in computer simulations in the field of agricultural engineering in recent years. DEM is not only widely used in the modeling, parameter calibration and simulation analysis of agricultural bulk materials [13,14], but it also plays a great role in the simulation analysis of stalk material characteristics. Kattenstroth et al. [15] constructed a stalk model based on the multi-sphere approach and bonded-sphere approach of DEM and then analyzed the effects of relevant parameters on the cutting quality by simulating the stalk-cutting process. Li et al. [16] constructed stalk and grain models in EDEM and then simulated and analyzed the motion and separation of stalks and grain in the airflow field by adopting computational fluid dynamics and the discrete element method (CFD–DEM). Xu et al. [17] constructed cucumber stalk models in EDEM and then simulated the stalk-crushing process. Ramirezgomez et al. [18] constructed the models of corn stalks, rice husks, and forage rape stems in EDEM and then calibrated the contact parameters for the density and models of blocky biomass fuel pressed by such materials. By combining the physical and simulation tests of the repose angle of corn stalk particles, Fang et al. [19] designed a Plackett–Burma test to screen for significant contact parameters and calibrated the static friction coefficient and rolling friction coefficient of particles using the response surface method. In order to obtain more accurate DEM simulation parameters in the compression process of the alfalfa stalk, Ma et al. [20] used the Plackett–Burman test and the Box–Behnken test design to calibrate the contact parameters of the alfalfa stalk based on the repose angle physical test. By combining the linear model of cohesion and the Hertz–Mindlin (no slip) contact model, Feng et al. [21] constructed a stalk model in EDEM, calibrated the simulation parameters of the stalk-adopting orthogonal test and analyzed the motion characteristics of the stalk model in the rotary drum. Liao et al. [22] measured the intrinsic parameters of forage rape stem by adopting physical tests and calibrated the discrete element contact parameters of forage rape stem particles based on DEM, with the repose angle of the stalk particles as the response value. The optimal combination of contact parameters was determined by adopting response surface analysis and optimization. Finally, the accuracy of the calibration parameters was verified by comparative tests. Zhang et al. [23] constructed the corn stalk model by using the Hertz–Mindlin with Bonding contact model in EDEM and calibrated the contact parameters between the corn stalk and its external working parts by combining the physical and simulation tests. Ma et al. [24] constructed the models of rice particles and stalks by using the Hertz–Mindlin contact model and used them to simulate the separation process of rice particles and impurities in the operation of a grain combined harvester. Based on the Hertz–Mindlin with Bonding contact model, Liu et al. [25] constructed a flexible wheat stalk model by DEM and simulated its bending behavior. Wang et al. [26] measured the intrinsic parameters and contact parameters of the citrus stalk by adopting physical tests, constructing the citrus stalk DEM model, and finally calibrating the bonding parameters of the model by adopting

the three-point bending test and shearing test. Zeng and Chen [27] constructed a simulation model of wheat straw by DEM without considering the bending and plastic deformations of wheat stalks, assigning the model's intrinsic parameters and contact parameters for use in the simulation experiment of stubble-free tillage. Guo et al. [28] constructed the tomato stalk model with an equal stalk diameter by using the multi-sphere approach and bonded-sphere approach in EDEM, assigning the model's basic contact parameters to simulate the mixing process. The above literature demonstrates the extensive research conducted by researchers on DEM-model construction, simulation parameters and bonding parameters calibration for various types of crop stalks. Among them, research on cotton stalks remains scarce. In addition, the shape and actual structure are quite different, as the stalk DEM model is mainly constructed by bonding multiple equal-diameter particle spheres, resulting in poor consistency between the simulation results and the actual test results.

For that reason, based on the biological structure characteristics of cotton stalks, the single distribution method is adopted to arrange the particle groups in the epidermis and internal tissues (xylem and pith) of the chopped cotton stalks [29], and the method of bonding elements of nonequal-diameter basic particles is adopted to construct a DEM model of cotton stalk particles in this paper. The repose angle of chopped cotton stalks was obtained by adopting the cylinder-lifting method, and the relevant parameters required for the simulation were obtained by physical tests. The contact parameters between cotton stalks were calibrated by adopting the repose angle simulation test, and the second-order response model between the repose angle and the contact parameters was established. The effect of the restitution coefficient, static friction coefficient and rolling friction coefficient between cotton stalks on the repose angle was obtained. On this basis, the optimal combination of contact parameters is obtained by taking the value of the repose angle physical test as the optimization target value, and the simulation verification is carried out. The objectives of this study are as follows: (1) To analyze the effect of the contact parameters between particles on the repose angle of chopped cotton stalks; (2) to obtain the DEM parameters of chopped cotton stalk particles; (3) to verify the accuracy of the calibrated parameters. The research provides a basis for the discrete element simulation study of cotton stalk motion in the separation process of cotton stalks and residual film and the subsequent gas–solid coupling simulation research.

## 2. Materials and Methods

### 2.1. Test Materials

The "Xinluzao No. 45" cotton from the cotton planting test field of Shihezi University in Xinjiang was selected as the test material and was sampled on 5 November 2021, with the moisture content of cotton stalks ranging from 22.03% to 36.16% and the diameter ranging from 5 mm to 15 mm. Cotton stalks without pests and bending damage were selected to ensure the experiment effect. In the mechanically-harvested residual film mixtures in the cotton field, the particle sizes of cotton stalks were different after being crushed by hammer-type crushing devices. Therefore, the test samples were prepared by adopting the methods of determining the repose angle of chopped stalks in references Fang et al. [19] and Sun et al. [30]. After the lateral branches were removed, the cotton stalks were chopped according to the length, ranging from 10 mm to 20 mm, and the particles at the stem-bud bulges and lateral branches were removed, as shown in Figure 1a. In terms of biological structure, the cotton stalk was divided into the epidermis, xylem and pith, as shown in Figure 1b.

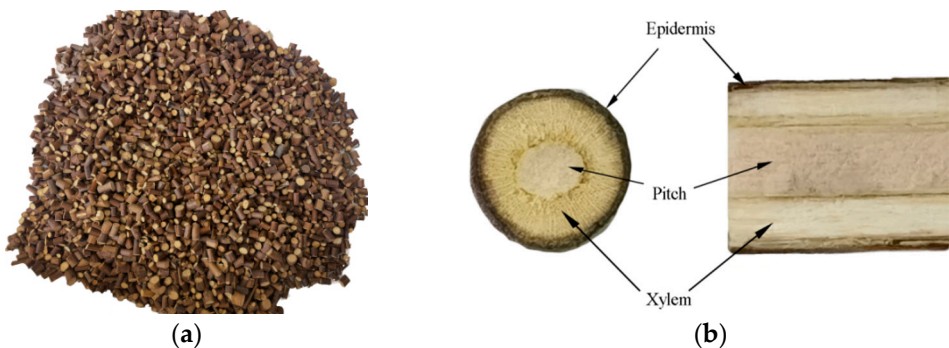

(a)                (b)

**Figure 1.** (**a**) Chopped cotton stalk particles; (**b**) biological structure of cotton stalk.

### 2.2. Simulation Contact Model Selection and Model Construction

2.2.1. Simulation Contact Model Selection

The DEM contact model of the particles in the simulation test had a significant effect on the test results. There is no adhesion between the cotton stalk particles in the simulation, so the effect of surface energy is ignored [31]. Meanwhile, it is assumed that the difference in the small overlaps between particles or between the particles and contact materials determines the changes in parameters, such as the displacement, force and velocity of particles in motion. According to Newton's Second Law of Motion, each particle model produces motion under the action of force and torque, and there is normal motion, tangential motion and rolling between particles. Based on the above assumptions, the Hertz–Mindlin (no slip), which is a default model in EDEM, was selected as the contact model of cotton stalk particles [32–34], as shown in Figure 2.

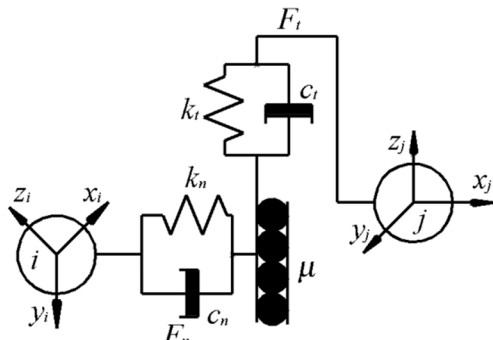

**Figure 2.** Schematic diagram of Hertz–Mindlin (no slip) contact model.

In this contact model, the relationship equations between the normal force ($F_n$) and tangential force ($F_t$) of particles and the overlaps $\delta_n$ and $\delta_t$ in their respective directions can be expressed as:

$$\begin{cases} F_n = \frac{4}{3}\delta_n^{\frac{3}{2}} E^* \sqrt{r^*} \\ F_t = -S_t \delta_t \end{cases} \tag{1}$$

where $E^*$ is Young's modulus, $r^*$ is the equivalent radius, $\delta_n$ is normal overlap, $\delta_t$ is tangential overlap, and $S_t$ is tangential stiffness. Where the $E^*$, $r^*$ and $S_t$ can be given by Equation (2):

$$\begin{cases} E^* = \frac{E_i + E_j}{E_i(1-v_j^2) + E_j(1-v_i^2)} \\ r^* = \frac{r_i r_j}{r_i + r_j} \\ S_t = 8G^* \sqrt{r^* \delta_n} \\ G^* = \frac{E^*}{2(1+\mu)} \end{cases} \tag{2}$$

where $E_i$ and $E_j$ are the Young's modulus of particles $i$ and $j$, $r_i$ and $r_j$ are the radii of particles $i$ and $j$, $\nu_i$ and $\nu_j$ are the Poisson's ratio of particles $i$ and $j$, $G^*$ is the equivalent shear modulus, and $\mu$ is Poisson's ratio.

Meanwhile, the normal damping force $F_n^d$ and tangential damping force $F_t^d$ of particles can be expressed as:

$$\begin{cases} F_n^d = -v_n \beta \sqrt{\dfrac{10 S_n m^*}{3}} \\ F_t^d = -v_t \beta \sqrt{\dfrac{10 S_t m^*}{3}} \end{cases} \tag{3}$$

where $m^*$ is the equivalent mass, $v_n$ and $v_t$ are the relative normal velocity and relative tangential velocity, and $\beta$ is the damping ratio. Where $m^*$, $S_n$ and $\beta$ can be given by Equation (4):

$$\begin{cases} \beta = \dfrac{\ln e}{\sqrt{\ln^2 e + \pi^2}} \\ m^* = \left( \dfrac{1}{m_i} + \dfrac{1}{m_j} \right)^{-1} \\ S_n = 2 E^* \sqrt{r^* \delta_n} \end{cases} \tag{4}$$

where $m_i$ and $m_j$ are the mass of particles $i$ and $j$, and $e$ is the coefficient of restitution.

The particles are inevitably affected by the rolling friction in motion; thus, the rolling friction can be explained by the torque $T_i$ on the contact surface [35], such as in Equation (5):

$$T_i = -\mu_r F_n R_i \omega_i \tag{5}$$

where $\mu_r$ is the coefficient of rolling friction, $R_i$ is the radius of particle $i$, and $\omega_i$ is the angular velocity.

### 2.2.2. Discrete Element Model Construction of Cotton Stalk Particles

In the simulation, a suitable cotton stalk particle model must be constructed. It should be ensured that the geometric and physical characteristics of cotton stalk particles in the DEM model match those of real ones. The complicated microstructure of cotton stalk particles makes it difficult to construct a DEM model that is exactly the same as the real structure. Therefore, by referring to the method of constructing the stalk DEM model by relevant scholars and thus combining the biological structure characteristics of cotton stalk, a single distribution method was adopted to arrange the particle groups in the epidermis and internal tissues (xylem and pith) of the chopped cotton stalks, and then the DEM model of cotton stalk particles was constructed by bonding the elements of nonequal-diameter basic particles [36,37], as shown in Figure 3. The same particle model was adopted for the xylem and pith of cotton stalks to make the calculation simpler and more efficient in the simulation [38]. The constructed cotton stalk particle model was 20 mm in height and 10 mm in diameter and filled with 612 particles. Among them, the radii of the spherical elements constituting the epidermis and internal tissues of the cotton stalk model were 1 mm and 2 mm, respectively, and their numbers are 494 and 108, respectively. In the simulation test, the diameter and height of the cotton stalk particle model were generated at random, ranging from 0.5 to 1 times the size of the basic cotton stalk model.

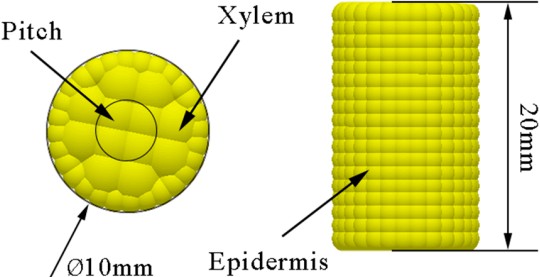

**Figure 3.** Basic simulation model of cotton stalk particles.

### 2.3. Experimental Determination of Some DEM Input Parameters

The accuracy of the DEM simulation model parameters directly affects the reliability of the simulation results. Because of the irregular shape of particles, the contact parameters between the particles measured by conventional methods are not accurate and have large errors, which cannot be directly used in the simulation test. For this reason, it is necessary to calibrate the contact parameters between the cotton stalk particles. In this paper, the internal parameters of the cotton stalk and its contact parameters with contact materials were measured by physical tests and set as fixed values. The same measurement method was adopted to obtain the range values of the contact parameters between particles so as to provide reference data for calibrating the contact parameters between particles.

### 2.3.1. Cotton Stalk Density

The ratio of material mass to its volume is defined as density. Referring to ASTM D854-10 [39], the volume of the cotton stalk was measured by the liquid immersion method [19]. The mass of the cotton stalk was measured by a JMB5003 electronic balance (measuring accuracy: 0.001 g; Jiming Weighting and Calibration Equipment Co., Ltd, Yuyao, Zhejiang, China) to determine its density. In order to reduce the test error and avoid the liquid from being absorbed by the cotton stalk, the test was completed in a very short time, and we carried out multiple sets of repeated tests to reduce the test error.

### 2.3.2. Elastic Modulus and Poisson's Ratio

The compression test was used to calculate the elastic modulus and Poisson's ratio of cotton stalks. Cylindrical cotton stalk particles, with a length of 20 mm and a diameter of 10 mm, were prepared, and uniaxial plate compression tests were conducted by using a universal material testing machine [22]; the compression speed was set to 5 mm/min, and the loading displacement was set to 4 mm. High-speed photography was used to compare and analyze the change rates of the height and diameter before and after compression. The test process was recorded by a FASTEC-TS4 high-speed camera (Fastec Imaging, San Diego, CA, USA, 510 fps), and the distance between the camera and the sample was 500 mm. The test was repeated 20 times, and the average value was calculated. The elastic modulus and Poisson's ratio of cotton stalks were calculated by Equation (6).

$$\begin{cases} E = \dfrac{F_0 l_0}{S \Delta l} \\ \nu = \left| \dfrac{\varepsilon_2}{\varepsilon_1} \right| = \dfrac{\Delta d l_0}{\Delta l d_0} \end{cases} \tag{6}$$

where $E$ is the elastic modulus (MPa), $\nu$ is Poisson's ratio, $F_0$ is the external force on the cotton stalk (N), $l_0$ is the initial length (mm), $S$ is the initial cross-sectional area (mm), $\Delta l$ is the length deformation (mm), $\varepsilon_1$ is the longitudinal strain (%); $\varepsilon_2$ is the transverse strain (%), $d_0$ is the initial diameter (mm), and $\Delta d$ is the diameter deformation (mm).

### 2.3.3. Coefficient of Restitution

The coefficient of restitution, which is defined as the ratio of the velocity of an object after a collision to that before the collision, reflects the ability of an object to recover to its original state after a collision. Most studies on the coefficient of restitution of agricultural materials equate irregular materials with spherical particles, thus simplifying the collision model and ignoring the effect of the material's shape on the coefficient of restitution. Therefore, this paper considers the effect of the shape and anisotropy of culm-like materials on the coefficient of restitution. According to the test method in references [40,41], the spatial velocity after a collision between the cotton stalk and the contact material is obtained

by combining high-speed photography, and the coefficient of restitution is calculated based on the principle of energy conservation, as shown in Equation (7):

$$e = \sqrt{\frac{\sum E_{k,o}}{\sum E_{k,i}}} = \sqrt{\frac{V_t^2 \left(57 + \frac{l^2}{r^2}\right)}{48 V_0^2}} \tag{7}$$

where $e$ is the coefficient of restitution, $\sum E_{k,i}$ is the total kinetic energy before the collision, $\sum E_{k,o}$ is the total kinetic energy after a collision, $V_t$ is the rebound velocity of the material after colliding (m/s), $V_0$ is the instantaneous approaching velocity before the material collides with the collision material (m/s), $r$ is the radius (mm), and $l$ is the length (mm).

2.3.4. Static Friction Coefficient

The static friction coefficient is the ratio of the maximum static friction force applied to an object to the positive pressure. The oblique plane sliding method is commonly used to measure the static friction coefficient [42,43]. The gravity of an object of mass, $m$, is decomposed into two forces: a force, $F$, parallel to the oblique plane and a force, $T$, perpendicular to the oblique plane. When the inclination angle, $\gamma$, of the oblique plane is less than the sliding critical angle, $F$ is less than the static friction force, $f$, between the object and the oblique plane, and the object remains static. With the increase of $\gamma$, $F$ increases. When $\gamma$ is greater than the sliding critical angle of the object, $F$ is greater than $f$ at this time, and the object will begin to slide along the oblique plane. Among them, the relationship between the static friction coefficient, $\mu_s$, and the inclination angle, $\gamma$, is shown in Equation (8):

$$\mu_s = \frac{f}{T} = mg \sin \gamma / (mg \cos \gamma) = \tan \gamma \tag{8}$$

The schematic diagram of the static friction coefficient measuring device is shown in Figure 4. During the test, one end of the lifting plate is connected with the lifting device of the universal testing machine by a rope, where the lifting device is adjusted to be in a horizontal position, and the contact material panel is fixed on the lifting plate at the same time. It should start the universal testing machine to make the lifting plate rise uniformly. The test process was recorded by a FASTEC-TS4 high-speed camera (Fastec Imaging, San Diego, CA, USA). The video was analyzed by ProAnalyst software to obtain the $\gamma$ value displayed by the electronic goniometer when the cotton stalk sample began to slide, and the static friction coefficient, μs, was calculated by Equation (8). The static friction coefficient measurement tests for each group were repeated 10 times, and the average was calculated.

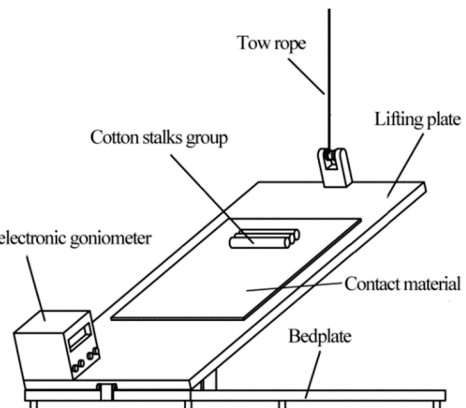

**Figure 4.** Schematic diagram of the static friction coefficient measuring device.

2.3.5. Rolling Friction Coefficient

According to the theory and experimental method in reference [44], high-speed photography was used to determine the rolling friction coefficient between the cotton stalks

and contact materials. Through theoretical analysis, there is a linear relationship between the ratio of energy loss and total energy in the process of particle rolling from the oblique plane and the tangent value of the oblique plane angle, and the slope is the rolling friction coefficient, as shown in Equation (9):

$$C_f = \frac{W_{fr}}{U} = \frac{U - E_k}{U} = \mu_r \cot \theta \tag{9}$$

where $C_f$ is the energy ratio of rolling friction loss (%), $U$ is the initial gravitational potential energy (J), $W_{fr}$ is the effort of friction force (J), $E_k$ is the kinetic energy at the end of particle motion (J), $\mu_r$ is the rolling friction coefficient, the radius (mm), and $\theta$ is the oblique plane angle (°).

In the test, the cotton stalk was placed on the oblique plane with an oblique angle of $\theta$ and a length of $L$ so that it could roll down from a certain height along the oblique plane. The initial gravitational potential energy of the cotton stalk is $U = mgL\sin\theta$. The instantaneous speed $v_s$ (m/s), when the cotton stalk moves to the bottom end of the oblique plane, is obtained by high-speed photography technology, and then the instantaneous kinetic energy of the cotton stalk is calculated as $E_k = 0.5\ mv_s^2$. In the test, the $\theta$ ranges from 15° to 50°, a total of 8 angle values are set, and each angle is repeated 10 times. The linear relationship between $C_f$ and $\cot\theta$ is fitted out by a linear regression equation, and the linear slope is the rolling friction coefficient.

### 2.4. Physical Test of Repose Angle and DEM Simulation Test
#### 2.4.1. Rolling Friction Coefficient

The cylinder-lifting method was applied for the repose angle test [45,46]. The diameter of the cylinder was set to be 4 to 5 times larger than the maximum length of the cotton stalk particles; thus, the diameter was set to 100 mm and the height to 200 mm. In order to ensure that the repose angle is only affected by the interaction of the contact parameters between cotton stalks, a circular chassis, which was formed by cotton-stalk particle paving, was used as the substrate (Figure 5a), and the universal testing machine was used to lift the cylinder upward at a speed of 50 mm·s$^{-1}$, making the cotton stalk a particle pile, as shown in Figure 5b.

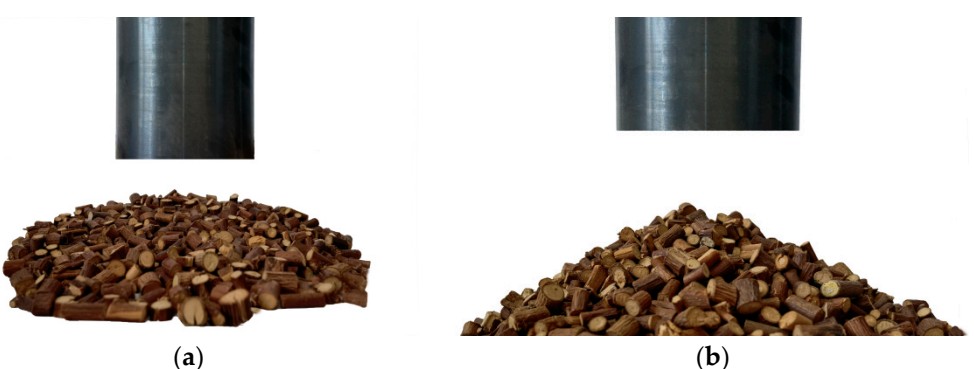

(**a**) (**b**)

**Figure 5.** Physical test of repose angle: (**a**) the circular chassis substrate; (**b**) stacking test results.

A high-definition camera was used in the test to record the boundary information of cotton stalk particle stacking. The MATLAB software was used for preprocessing, gray processing, hole filling and binarization. The boundary information of the repose angle of cotton stalks was extracted. Finally, the least-squares method was used to fit the extracted stacking boundary linearly, so the slope of the fitted line was the tangent of the repose angle. The image processing is shown in Figure 6. The repose angle physical test was conducted 20 times. The average repose angle of the cotton stalk particles is 26.45°, and the standard deviation is 0.57°.

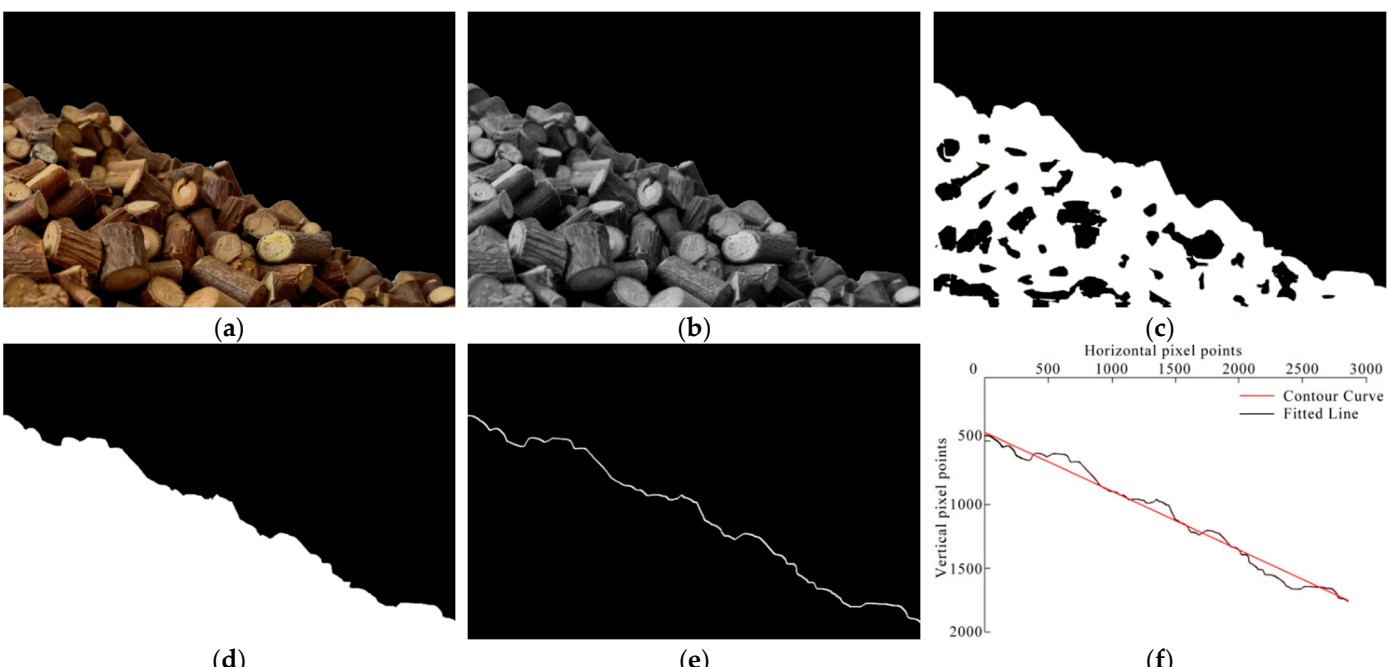

**Figure 6.** Image processing results: (**a**) preprocessing; (**b**) gray-scale processing; (**c**) hole filling; (**d**) binarization; (**e**) boundary extraction; (**f**) fitting of one-sided repose angle.

### 2.4.2. DEM Simulation Test

Based on the physical test equipment and discrete element model, the simulation test of the repose angle was conducted in EDEM. The cylinder model was constructed according to the actual size in Solidworks and imported into EDEM, and a particle factory was set at the top of it. In the simulation test, the time step was set to 20% of that of Rayleigh, and the total simulation time was set to 5 s, in which the time of generating the cotton stalk particles model and system stability was 1.8 s. The vertical upward lifting speed of the cylinder was set to 50 mm·s$^{-1}$, and the cotton stalk particles slowly overflowed from the bottom of the cylinder, which finally formed a stable particle pile on the bottom plate, as shown in Figure 7. After the simulation test of the repose angle, the particle pile images under the +X and +Y views were collected. The image processing method in Section 2.4.1 was used to extract and fit the boundary of the repose angle, and the average value was taken as the simulation repose angle.

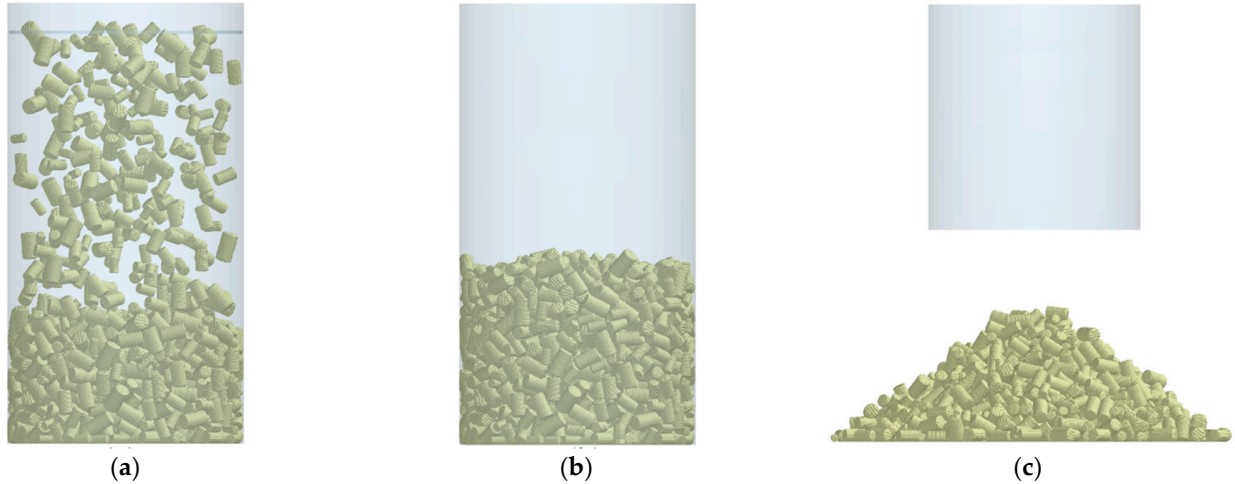

**Figure 7.** Simulation test process of repose angle: (**a**) particle filling; (**b**) system reaches stability; (**c**) heap completion.

### 2.5. Determination of DEM Input Parameters and Test Scheme

2.5.1. Determination of DEM Input Parameters

By referring to the literature and determining the input parameters in Section 2.3, the intrinsic parameters of cotton stalks and steel, and the contact parameters between them, were obtained. Because the intrinsic parameters have little effect on the repose angle, and the contact parameters between the cotton stalks and the steel can be determined by tests, those parameters can be set as fixed values. The input parameters under the simulation test are shown in Table 1.

**Table 1.** The input parameters of the simulation test.

| DEM Parameter | Parameter Value | DEM Parameter | Parameter Value |
|---|---|---|---|
| Density of Cotton Stalk (g·cm$^{-3}$) | 0.326 | Cotton Stalk-Steel coefficient of restitution | 0.56 |
| Elastic modulus of Cotton Stalk (Pa) | $4.34 \times 10^9$ | Cotton Stalk-Steel static friction coefficient | 0.62 |
| Poisson's ratio of Cotton Stalk | 0.35 | Cotton Stalk-Steel rolling friction coefficient | 0.16 |
| Density of Steel (g·cm$^{-3}$) | 7.85 | * Cotton Stalk-Cotton stalk coefficient of restitution | 0.16~0.48 |
| Elastic modulus of Steel (Pa) | $2.06 \times 10^{11}$ | * Cotton Stalk-Cotton stalk static friction coefficient | 0.45~0.65 |
| Poisson's ratio of Steel | 0.30 | * Cotton Stalk-Cotton stalk rolling friction coefficient | 0.10~0.20 |

Note: * is the parameter to be calibrated.

2.5.2. Response Surface Test Design

The central composite design (CCD), as a design of the experiment commonly used in response surface methodology (RSM), is facilitated to analyze the effects of various factors on the response value and optimize the results. The CCD module in the Design Expert software is used to encode the contact parameters between cotton stalks: the coefficient of restitution ($X_1$), static friction coefficient ($X_2$) and rolling friction coefficient ($X_3$). The coding of the influencing factors and the setting of horizontal values in the simulation test of the repose angle of cotton stalk particles are shown in Table 2.

**Table 2.** Influencing factors and their level values of simulation test of repose angle.

| Level | $X_1$ | $X_2$ | $X_3$ |
|---|---|---|---|
| −1.68 | 0.05 | 0.38 | 0.07 |
| −1 | 0.16 | 0.45 | 0.1 |
| 0 | 0.32 | 0.55 | 0.15 |
| 1 | 0.48 | 0.65 | 0.20 |
| 1.68 | 0.59 | 0.71 | 0.23 |

According to the influencing factors and their horizontal values set in Table 2, the simulated repose angle ($Y$) of cotton stalk particles was taken as the response value, and the three-factor five-level central composite test was designed by Design Expert software 8.0.6 (Stat Ease Inc., Minneapolis, MN, USA). The second-order response model between the response values and the contact parameters was constructed to obtain the effect laws of the coefficient of restitution, the static friction coefficient and the rolling friction coefficient on the repose angle. The test scheme is shown in Table 3.

**Table 3.** The scheme and results of repose angle simulation tests.

| Test No. | Coding | | | Response Value | Test No. | Coding | | | Response Value |
|---|---|---|---|---|---|---|---|---|---|
| | $X_1$ | $X_2$ | $X_3$ | $Y$ (°) | | $X_1$ | $X_2$ | $X_3$ | $Y$ (°) |
| 1 | −1 | −1 | −1 | 25.52 ± 1.32 | 11 | 0 | −1.68 | 0 | 25.47 ± 1.23 |
| 2 | 1 | −1 | −1 | 23.43 ± 2.51 | 12 | 0 | 1.68 | 0 | 28.36 ± 1.14 |
| 3 | −1 | 1 | −1 | 25.28 ± 1.36 | 13 | 0 | 0 | −1.68 | 23.69 ± 1.46 |
| 4 | 1 | 1 | −1 | 25.64 ± 1.85 | 14 | 0 | 0 | 1.68 | 31.52 ± 2.65 |
| 5 | −1 | −1 | 1 | 27.96 ± 1.34 | 15 | 0 | 0 | 0 | 25.49 ± 1.28 |
| 6 | 1 | −1 | 1 | 30.3 ± 2.84 | 16 | 0 | 0 | 0 | 25.02 ± 1.25 |
| 7 | −1 | 1 | 1 | 25.68 ± 3.01 | 17 | 0 | 0 | 0 | 25.18 ± 1.64 |
| 8 | 1 | 1 | 1 | 33.76 ± 2.58 | 18 | 0 | 0 | 0 | 25.55 ± 0.87 |
| 9 | −1.68 | 0 | 0 | 25.61 ± 1.47 | 19 | 0 | 0 | 0 | 26.51 ± 1.84 |
| 10 | 1.68 | 0 | 0 | 28.26 ± 1.65 | 20 | 0 | 0 | 0 | 24.43 ± 1.91 |

Note: ± is the standard deviation.

## 2.6. Optimum Condition and Validation

In order to obtain the optimal input parameter combination of contact parameters between cotton stalk particles, by combining with the boundary conditions of the contact parameters in Section 2.5.1, the optimization module in the Design Expert software was utilized to optimize the second-order response model, with the physical test value of the repose angle set as the target values. Taking this parameter combination as the simulation input parameters, the simulation test of the repose angle was conducted. By comparing the relative error of the repose angle in the simulation test and the physical test, the accuracy and reliability of the optimal combination parameters were verified.

## 3. Results and Discussion

### 3.1. Analysis of the Simulation Test Result of the Repose Angle

3.1.1. ANOVA and Model Construction

The simulation test was conducted according to the test scheme in Table 3, and the simulated repose angles of the cotton stalk particles under different parameter combinations were obtained, as shown in Table 3. The simulation test results were analyzed by analysis of variance (ANOVA) through the data processing module of the Design Expert software, as shown in Table 4.

**Table 4.** ANOVA of the test results.

| Source | Sum of Squares | Mean Square | *F* Value | *p*-Value |
|---|---|---|---|---|
| *Model* | 129.06 | 14.34 | 29.88 | <0.0001 ** |
| $X_1$ | 12.66 | 12.66 | 26.37 | 0.0004 ** |
| $X_2$ | 4.70 | 4.70 | 9.79 | 0.0107 * |
| $X_3$ | 70.36 | 70.36 | 146.59 | <0.0001 ** |
| $X_1 X_2$ | 8.38 | 8.38 | 17.47 | 0.0019 ** |
| $X_1 X_3$ | 18.45 | 18.45 | 38.44 | 0.0001 ** |
| $X_2 X_3$ | 0.078 | 0.078 | 0.16 | 0.6953 |
| $X_1{}^2$ | 4.24 | 4.24 | 8.83 | 0.014 * |
| $X_2{}^2$ | 4.13 | 4.13 | 8.60 | 0.015 * |
| $X_3{}^2$ | 8.75 | 8.75 | 18.22 | 0.0016 ** |
| Residual | 4.80 | 0.48 | | |
| Lack of Fit | 2.41 | 0.48 | 1.01 | 0.4959 |
| Pure Error | 2.39 | 0.48 | | |
| Cor Total | 133.86 | | | |
| Adeq Precision | 22.04 | | | |
| $R^2 = 0.9641$, $R^2{}_{adj} = 0.9319$, $R^2{}_{Pred} = 0.8216$, C.V = 2.60% | | | | |

Note: ** = extremely significant factor ($p \leq 0.01$); * = significant factor ($0.01 < p \leq 0.05$); NS = not significant factor ($p > 0.05$).

Through the regression analysis of variance on the test results, if the insignificant influencing factor in the second-order response model was eliminated, the response model between the contact parameters and the repose angle was obtained, as shown in Equation (10):

$$Y = 57.52 - 71.21X_1 - 70.51X_2 - 97.98X_3 + 63.98X_1X_2 + 189.84X_1X_3$$
$$-19.75X_2X_3 + 21.18X_1{}^2 + 53.51X_2{}^2 + 311.63X_3{}^2 \tag{10}$$

The results of ANOVA are shown in Table 4. The *F* value (29.88) and *p*-value ($p < 0.0001$) of the model indicated that the model was extremely significant within the 95% confidence interval. The *F* value of the Lack of Fit term is 1.01, indicating that the Lack of Fit term is not significant relative to the pure error. The *p*-value of the Lack of Fit term is 0.4959, which is far greater than 0.05 and extremely insignificant. The coefficient of variation, *C.V* (2.60%), is low, indicating that the second-order response model between the contact parameters and the repose angle obtained by ANOVA is reliable.

In addition, the determination coefficient ($R^2$) and the adjusted determination coefficient ($R^2_{adj}$) of the second-order response model were 0.9641 and 0.9319, respectively, indicating a high fit of the model to the experimental data. The difference between $R^2_{adj}$ (0.9319) and the predicted determination coefficient $R^2_{Pred}$ (0.8216) was small (less than 0.2), and the SNR (Adeq precision) was 22.04%, indicating that the test factors have a high degree of interpretation for the response values. The result is further demonstrated by the fact that the scatter of predicted values is very close to the straight line from Figure 8 or the diagram of the normal plot of residuals.

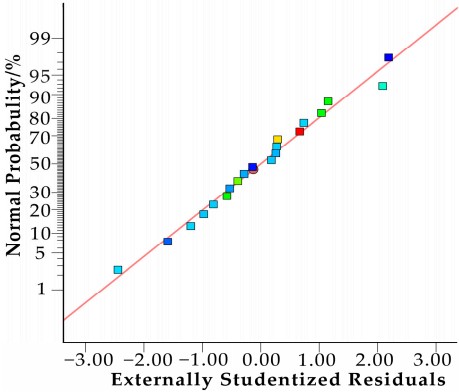

**Figure 8.** Normal plot of residuals.

3.1.2. Single-Factor Effect Analysis

As shown in Table 4, the single-factor terms $X_1$, $X_3$, and secondary term $X_3{}^2$ have an extremely significant ($p < 0.01$) effect on the repose angle. The *p*-values of $X_2$, $X_1{}^2$ and $X_2{}^2$ are all from 0.01 to 0.05, so they are the significant effect factors of the repose angle. In order to analyze the effect trend of the test factors $X_1$, $X_2$ and $X_3$, respectively, on the repose angle ($Y$) more intuitively, any two factors in the second-order response model are set to a central level value of 0, and the effect law of the single-factor on the repose angle is obtained, respectively, as shown in Figure 9.

It can be seen in Figure 9 that when $X_1$ changes from a low level ($-1.68$) to a high level (1.68), that is, when the coefficient of restitution gradually increases from 0.05 to 0.59, the repose angle decreases first and then increases. When the level value of $X_1$ was $-0.899$, the minimum value of the repose angle was 24.94°. The effect law of the single-factor term $X_2$ on the repose angle is the same as that of $X_1$. When the static friction coefficient increases gradually from 0.38 to 0.71, the value of the repose angle also decreases first and then increases. As the level value of $X_3$ changes from a low level ($-1.68$) to a high level (1.68), the repose angle gradually increases; that is, when the rolling friction coefficient gradually increases from 0.07 to 0.23, the repose angle gradually increases. In addition, with the level

value of $X_3$ changing from $-1.68$ to $-1$, the repose angle increased slowly, and with the level value of $X_3$ changing from $-1$ to $1.68$, the repose angle increased rapidly.

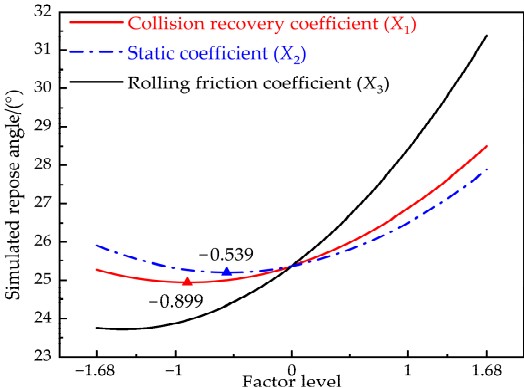

**Figure 9.** Single-factor response diagram.

### 3.1.3. Interaction Effect Analysis

As can be seen from the results of ANOVA in Table 4, among the interaction terms, the *p*-values of $X_1X_2$ and $X_1X_3$ were far less than 0.01, so $X_1X_2$ and $X_1X_3$ have an extremely significant effect on the repose angle. Because the *p*-value of $X_2X_3$ was far greater than 0.05, it showed a non-significant effect on the repose angle. To research the effect of the interaction terms on the repose angle, this paper plotted the response surface of the interaction terms $X_1X_2$ and $X_1X_3$ on the interaction of the repose angle and analyzed their effect law on the repose angle. The response surface is shown in Figure 10.

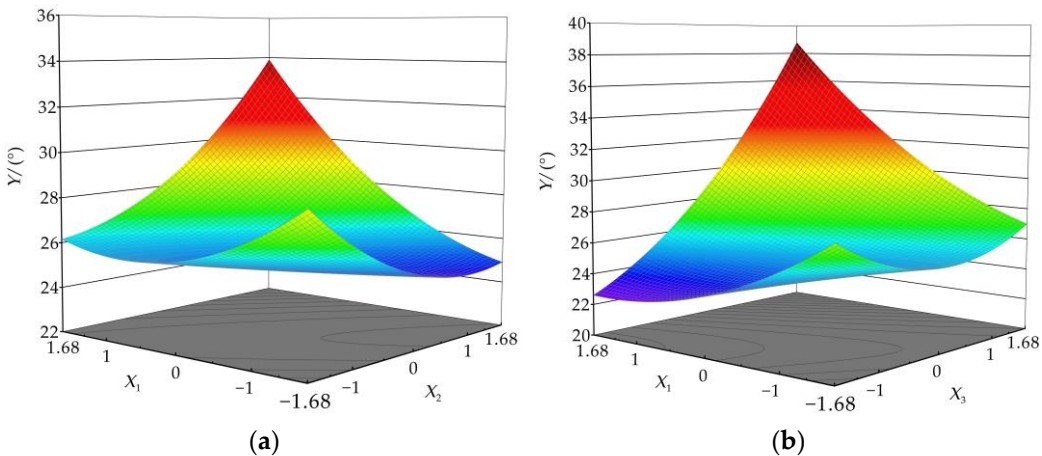

**Figure 10.** Influence of interaction terms on repose angle: (**a**) interaction term $X_1X_2$; (**b**) interaction term $X_1X_3$.

1. Effect of interaction term $X_1X_2$ on the repose angle

When the level value of single-factor term $X_3$ is at the center level 0, the effect law of interaction term $X_1X_2$ on the repose angle is shown in Figure 10a. When the level value of $X_1$ is at $-1.68$ and the level value of $X_2$ increases from $-1.68$ to $1.68$, the repose angle decreases first and then increases. This change is the same as the effect law of the single-factor term $X_2$ on the repose angle. When the level value of $X_1$ is at $1.68$ and the level value of $X_2$ increases from $-1.68$ to $1.68$, the repose angle increases sharply. This change differs from the effect law of the single-factor $X_2$ on the repose angle. Among the two-factor interaction terms $X_1X_2$, the effect law of single-factor $X_2$ on the repose angle is the same as that of $X_1$. However, the slope of the $X_1$ response surface curve is steeper than the $X_2$ direction, indicating that the effect of $X_1$ on the repose angle is more significant than that of $X_2$.

2.    Effect of interaction term $X_1X_3$ on repose angle

When the level value of single-factor term $X_2$ is at the center level 0, the effect law of interaction term $X_1X_3$ on the repose angle is shown in Figure 10b. When the level value of $X_1$ is at $-1.68$ and the level value of $X_3$ increases from $-1.68$ to 1.68, the repose angle decreases first and then increases. This change differs from the effect law of the single-factor term $X_3$ on the repose angle. When the level value of $X_1$ is at 1.68 and the level value of $X_3$ increases from $-1.68$ to 1.68, the repose angle increases sharply. This change is the same as the effect law of the single-factor $X_3$ on the repose angle. Among the two-factor interaction terms $X_1X_3$, the effect law of the single-factor $X_3$ on the repose angle is the same as that of $X_1$. However, the slope of the $X_3$ response surface curve is steeper than the $X_1$ direction, indicating that the effect of $X_3$ on the repose angle is more significant than that of $X_1$. By comparing Figure 10a,b, it can be concluded that the effect order of the single-factor on the repose angle is $X_3 > X_1 > X_2$, which is consistent with the ANOVA results.

*3.2. Determination of Optimal Parameter Combinations and Verification*

By taking the physical test values of repose angle ($Y$) as the optimization target values and $X_1$, $X_2$ and $X_3$ as the optimization values, the optimization module in Design Expert software was used to solve the second-order response model (i.e., Equation (10)) to obtain the optimal parameter combination. The single-objective and multi-variable optimization method was used to establish the objective and constraint functions. The mathematical model of parameter optimization is as follows:

$$\begin{cases} Y(X_1, X_2, X_3) = 26.45° \\ \begin{cases} 0.16 \leq X_1 \leq 0.48 \\ 0.45 \leq X_2 \leq 0.65 \\ 0.1 \leq X_3 \leq 0.2 \end{cases} \end{cases} \tag{11}$$

The optimal combination of contact parameters after calibration is $X_1 = 0.45$, $X_2 = 0.47$ and $X_3 = 0.16$. In order to verify the accuracy of the optimal parameter combination, the simulation test of the repose angle of cotton stalk particle stacking was conducted after entering the obtained contact parameters into EDEM.

The simulation test was repeated five times, and the simulated values of the repose angle were $26.91°$, $25.90°$, $26.15°$, $26.43°$ and $26.40°$, respectively. The relative errors between the simulated value of the repose angle and the physical test value were 1.74%, 2.08%, 1.13%, 0.08%, and 0.19%, respectively. In the simulation test, the average value of the repose angle was $26.36°$, and the average relative error was 1.04%. There was no significant difference between the simulated values and the physical test values, indicating that the optimal parameter combination was accurate and reliable, further verifying that the second-order response model between the coefficient of restitution, static friction coefficient, rolling friction coefficient and repose angle is suitable for the calibration of cotton stalk contact parameters.

**4. Conclusions**

The DEM simulation was used to study the repose angle of chopped cotton stalk particles. The method can be used as an application to analyze the effects of DEM parameters on predicting the repose angle. The following conclusion can be drawn from this study:

(1)    The contact parameters between the cotton stalk particles and the contact material (steel) were measured by physical tests. The coefficient of restitution was 0.56, the static friction coefficient was 0.62, and the rolling friction coefficient was 0.16. As for the range of contact parameters between the cotton stalk particles, the coefficient of restitution ranged from 0.16 to 0.48, the static friction coefficient ranged from 0.45 to 0.65, and the rolling friction coefficient ranged from 0.10 to 0.20. The cylinder-lifting method was applied to test the repose angle of chopped cotton stalks, and the average

value (26.45°) and the standard deviation (0.57°) of the repose angle of cotton stalk particles were obtained.

(2) A central composite design test on the response surface methodology was constructed to conduct the simulation test of the repose angle, the second-order response model between contact parameters and repose angle. According to the results of ANOVA, all the figures indicate that the test factors had a high interpretation of the response value.

(3) By analyzing the effects of single-factor and interaction factors on the repose angle, the extremely significant factors affecting the repose angle were the coefficient of restitution and rolling friction coefficient, while the static friction coefficient was the most significant factor. The coefficient of restitution interacted with the static friction coefficient and the rolling friction coefficient on the repose angle, and they had a significant effect on the repose angle.

(4) The optimal combination of contact parameters was determined as follows: the coefficient of restitution was 0.45, the static friction coefficient was 0.47, and the rolling friction coefficient was 0.16. The verification test showed no significant difference between the simulated and physical test values, and the average relative error was 1.04%, indicating that the simulated values agreed well with the physical test values, verifying the authenticity of the simulation test and the reliability of the optimal combination of simulation parameters. The research provides a basis for the discrete element simulation study of cotton stalk motion in the separation process of cotton stalks and residual film and thus could be used for subsequent gas–solid coupling simulation research.

**Author Contributions:** Conceptualization, methodology, formal analysis, writing—review and editing, writing—original draft preparation, B.Z. and Z.K.; supervision, X.C.; funding acquisition, Z.K.; investigation, H.M.; software, X.W.; data curation, R.L. All authors have read and agreed to the published version of the manuscript.

**Funding:** This research was funded by the National Natural Science Foundation of China, grant number: 52065058; the Graduate Education Innovation Project of Xinjiang Uygur Autonomous Region, grant number: XJ2022G085; and the Open Fund of Jiangsu Province and Education Ministry Co-sponsored Synergistic Innovation Center of Modern Agricultural Equipment, Grant No. XTCX2006.

**Institutional Review Board Statement:** Not applicable.

**Data Availability Statement:** The data presented in this study are available on request from the corresponding author.

**Conflicts of Interest:** The authors declare no conflict of interest.

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
