# Peer review of "Calibration and Test of Contact Parameters between Chopped Cotton Stalks Using Response Surface Methodology"

_agriculture, doi:10.3390/agriculture12111851_

Round 1

Reviewer 1 Report

The research is good, but the paper is not handled properly. The manuscript has some flaws that have to be addressed. The suggestions and comments are given in the attached file

Author Response

Point 1: Avoid using abbreviation in the abstract.

Response 1: Thank you for your valuable comments and suggestions. According to your suggestion, it has been revised in the abstract.

Point 2: Keywords are generic try to add more or improve keywords.

Response 2: Thank you for your valuable comments and suggestions. Keywords has been modified as: Discrete element method; Cotton stalk particles; Contact parameters; Calibration; Response surface methodology.

Point 3: Try to make this new paragraph and try to concise the 2nd paragraph.

Response 3: Thank you for your valuable comments and suggestions. According to your suggestion, it has been modified in the paper.

Point 4: Novelty and objective of this study must write in separate paragraph.

Response 4: Thank you for your valuable comments and suggestions. According to your suggestion, it has been modified in the paper.

Point 5: Add some lines about application or benefits of this study?

Response 5: Thank you for your valuable comments and suggestions. At the end of this paragraph, we added the sentence “The research provide a basis for the discrete element simulation study of cotton stalk motion in the separation process of cotton stalks and residual film and the subsequent gas-solid coupling simulation research”.

Point 6: Give reference of these equation for simulation and reference any researcher measured with these equipments.

Response 6: Thank you for your valuable comments and suggestions. We have added relevent references in this paper.

Point 7: Mention shutter speed, frames per sec and distance of lens from object. Also mention model and accuracy of the camera

Response 7: Thank you for your valuable comments and suggestions. According to your suggestion has been revised in the text. Because the whole test process was recorded, the shutter speedwas not set.

Point 8: R should in small letter.

Response 8: Thank you for your valuable comments and suggestions. According to your suggestion, it has been revised in the paper.

Point 9: Out side the page arrange it properly

Response 9: Thank you for your valuable comments and suggestions. According to your suggestion, we have adjusted the pictures and tables.

Point 10: Only P and F should be reported instead of whole ANOVA table.

Response 10: Thank you for your valuable comments and suggestions. Although the values of 'Sum of Squares' and the 'Mean Square' were not interpreted in detail during the analysis, we retained them in order to allow readers to observe the test results more intuitively.

Point 11: Try to explain your results of ANOVA in the form of research finding not explain the table as it is.

Response 11: Thank you for your valuable comments and suggestions. Our idea was to obtain the second-order response model between the contact parameters and the angle of repose by performing ANOVA on the test results. The reliability of the second-order response model was verified by interpreting the results of ANOVA. In the single-factor analysis and interaction effect analysis, the variance results were analyzed in the form of research results.

Point 12: Improve the figure looks like direct copy from software. Make it more visible and neat.

Response 12: Thank you for your valuable comments and suggestions. According to your suggestion, we have adjusted the figure.

Point 13: What are X1 and X2.

Response 13: Thank you for your valuable comments and suggestions. X1 and X2 were labeled in Section 2.5.2. X1 represented the coefficient of restitution of between cotton stalks and X2 represented the static friction coefficient between cotton stalks.

Point 14: Not understand. “Among the two-factor interaction termsX1X2, the effect law of single-factor X2 on the repose angle is the same as that of X1. However, the slope of the X1 response surface curve is steeper than the X2 direction, indicating that the effect of X1 on the repose angle is more significant than that of X1.”

Response 14: Thank you for your valuable comments and suggestions. Through the ANOVA, the interaction term X1X2 was extremely significant effect on the simulated  repose angle, but the influence of single-factor on the simulated repose angle was not exactly the same. By observing Fig. 10a, the influence of X1 and X2 on the simulated repose angle in the interaction term was obtained was obtained, and the differences between the two are compared. 

Point 15: Results are explained on the basis of ANOVA you must support your results mostly with research what are best results and how?

Response 15: Thank you for your valuable comments and suggestions. Through the ANOVA, we obtained the single-factor and interaction terms that have a significant effect on the repose angle, and constructed a second-order response model between the contact parameters and the repose angle, and the model does not contain the insignificant factors in the ANOVA. When optimizing and solving the optimal contact parameters, the second-order response model is constructed by ANOVA, and the results can support our conclusion.

Point 16: Conclusions should be concise and write in the form of point.

Response 16: Thank you for your valuable comments and suggestions. According to your suggestion, the conclusions have been simplified and written in the form of point.

Point 17: Add some future perspective at the end of the conclusions section.

Response 17: Thank you for your valuable comments and suggestions. According to your suggestion, it has been modified in the paper.

Reviewer 2 Report

1In cotton stalk density measurement, the volume of the cotton stalk was measured by the liquid immersion method. What kind of liquid was used? Would the liquid be absorbed by the cotton stalk when the volume was measured? This affects the volume measurement of the stalk.

2When Poisson's ratio of the cotton stalk was measured, the stalk was compressed by 4 mm. Was the cotton stalk still in the elastic stage at this point?

3Figure 4 shows a schematic diagram of the static friction coefficient measuring device. From the figure, did the stalk roll downward as it was being measured? Is this measurement of the rolling friction coefficient?

4Are the contact parameters between the cotton stalk cross-section and the steel plate measured? Is there a difference between this parameter and the contact parameter of the cotton stalk of the bast-steel plate? Is it reasonable to use the contact parameters of the cotton stalk of the bast-steel plate to replace the contact parameters of the cross-section-steel plate?

5When the contact parameters of the cotton stalk-stalk were measured, were only the contact parameters of bast-bast measured? Are the contact parameters between the cross-section of the cotton stalk and the bast measured? Are the contact parameters between the cotton stalk cross-section and the cross-section measured? Is there a difference between these contact parameters? Is it reasonable to use the contact parameters of cotton stalk phloem-phloem to replace these two contact parameters?

6A universal testing machine was used to lift a lifting plate at a constant speed to measure the coefficient of static friction. When the lifting plate was lifted, the tow rope would tilt and stretch the lifting plate. Would the device be dragged forward by the universal testing machine as it continued to rise vertically and at a constant speed?

7Check the format of Tables 1 and 3 for problems.

8X1, X2 and X3 are not clearly expressed. Is it the contact parameter of the cotton stalk-cotton stalk or the contact parameter of the cotton stalk-steel plate?

Author Response

Point 1: In cotton stalk density measurement, the volume of the cotton stalk was measured by the liquid immersion method. What kind of liquid was used? Would the liquid be absorbed by the cotton stalk when the volume was measured? This affects the volume measurement of the stalk.

Response 1: Thank you for your valuable comments and suggestions. The liquid immersion method is a relatively accurate measurement of cotton stalk density, enabling the liquid to reach all parts of the material. The liquid used to measure the volume of cotton stalk is pure water. In order to reduce the test error and avoid the liquid being absorbed by the cotton stalk, the test was completed in a very short time, and we carried out multiple sets of repeated tests to reduce the test error.

Point 2: When Poisson's ratio of the cotton stalk was measured, the stalk was compressed by 4 mm. Was the cotton stalk still in the elastic stage at this point?

Response 2: Thank you for your valuable comments and suggestions. When measuring the Elastic modulus and Poisson's ratio, in order to avoid friction between the end face of the sample and the supporting surface of the testing machine due to the expansion of the sample, the sample was selected as a length of 20 mm and a diameter of 10 mm. According to the literature (Liao, Y.; Wang, Z.; Liao, Q.; Wan, X.; Zhou, Y.; Liang, F. Calibration of discrete Element Model Parameters of Forage Rape Stalk at Early Pod Stage. Trans. Chin. Soc. Agricult. Machin. 2020, 51, 236–243.)and the actual test, it was found that when the cotton stalk was compressed by 4mm, it was not in the elastic stage.

Point 3: Figure 4 shows a schematic diagram of the static friction coefficient measuring device. From the figure, did the stalk roll downward as it was being measured? Is this measurement of the rolling friction coefficient?

Response 3: Thank you for your valuable comments and suggestions. Figure 4 shows a schematic diagram of the static friction coefficient measuring device. When measuring static friction coefficient, in order to avoid the rolling of a single cotton stalk, three cotton stalks adhered together. Referring to the measurement method of rolling friction coefficient in literature[44] ( Cui, T.; Liu, J.; Yang, L.; Zhang, D.; Zhang, R.; Lan, W. Experiment and simulation of rolling friction characteristic of corn seed based on high-speed photography. Trans. Chin. Soc. Agricult. Eng. 2013, 29, 34–41.), we used the device shown in Figure 4 to carry out the rolling friction coefficient measurement test.

Point 4: Are the contact parameters between the cotton stalk cross-section and the steel plate measured? Is there a difference between this parameter and the contact parameter of the cotton stalk of the bast-steel plate? Is it reasonable to use the contact parameters of the cotton stalk of the bast-steel plate to replace the contact parameters of the cross-section-steel plate?

Response 4: Thank you for your valuable comments and suggestions. Considering that the main region of the contact between the cotton stalk and the steel plate is the cotton stalk epidermis, and in order to simplify the parameter setting and improve the operation efficiency, the contact parameters between the cotton stalk cross-section and the steel plate are not measured in this paper. In the subsequent optimization study of cotton stalk model construction, we will measure this parameter and study its influence on the test results.

Point 5: When the contact parameters of the cotton stalk-stalk were measured, were only the contact parameters of bast-bast measured? Are the contact parameters between the cross-section of the cotton stalk and the bast measured? Are the contact parameters between the cotton stalk cross-section and the cross-section measured? Is there a difference between these contact parameters? Is it reasonable to use the contact parameters of cotton stalk phloem-phloem to replace these two contact parameters?

Response 5: Thank you for your valuable comments and suggestions. In the EDEM simulation test, it is impossible to construct a model that is completely consistent with the actual cotton stalk, and the parameters cannot be set to simultaneously characterize the difference between the cotton stalk phloem-phloem, the cross-section and the cross-section of the cotton stalk, and the cross-section and the phloem of the cotton stalk. Therefore, only the contact parameter values between the cotton stalk phloem and the phloem were measured, and the model was simplified. In the subsequent optimization study of cotton stalk model construction, we will measure different contact parameters and study their influence on the test results.

Point 6: A universal testing machine was used to lift a lifting plate at a constant speed to measure the coefficient of static friction. When the lifting plate was lifted, the tow rope would tilt and stretch the lifting plate. Would the device be dragged forward by the universal testing machine as it continued to rise vertically and at a constant speed?

Response 6: Thank you for your valuable comments and suggestions. We also found this problem in the pre-experiment, so we fixed the bedplate and kept the measuring device unable to be dragged during the experiment.

Point 7: Check the format of Tables 1 and 3 for problems.

Response 7: Thank you for your valuable comments and suggestions. We have adjusted the tables.

Point 8: X1, X2 and X3 are not clearly expressed. Is it the contact parameter of the cotton stalk-cotton stalk or the contact parameter of the cotton stalk-steel plate?

Response 8: Thank you for your valuable comments and suggestions. We have modified and indicated the meaning of these symbols in the paper.

Reviewer 3 Report

In order to using Discrete Element Method (DEM) to analyzing the force and movement during the separation of cotton stalk from residual film, this paper proposed a DEM model of cotton stalk particles, and determined key contact parameters between chopped cotton stalks by response surface.

The problems that this paper studied originate from engineering practice, and the research methods and conclusions are instructive to the engineering practice.

However, in my opinion, this paper has the following problems:

(1) The second-order response model has its own minimum. Finding out a parameters combination to making the response value equal to a repose angle of chopped cotton stalks that measured by physical tests, is not a problem of optimization, but a problem of solving equations. And, if the minimum of response surface is less than the measured repose angle, the solution is an isoline; if the minimum of response surface is greater than the measured repose angle, there is no solution. I think the description of the problem in formula (11) is inaccurate and incomplete.

(2) The test method of regression equation is used to verify the correctness of the response surface, but since the test data are all from simulation, how does the random error generate? Analysis of variance cannot be used without random error.

I think this article may accept after giving supplementary explanations to the above questions.

Author Response

Point 1: The second-order response model has its own minimum. Finding out a parameters combination to making the response value equal to a repose angle of chopped cotton stalks that measured by physical tests, is not a problem of optimization, but a problem of solving equations. And, if the minimum of response surface is less than the measured repose angle, the solution is an isoline; if the minimum of response surface is greater than the measured repose angle, there is no solution. I think the description of the problem in formula (11) is inaccurate and incomplete.

Response 1: Thank you for your valuable comments and suggestions. Through the regression analysis of variance on the test results,the response model between the contact parameters and the repose angle was obtained, maximum and minimum values are available within the selected contact parameter simulation range. The response surface is solved by taking the physical test value of the repose angle as the target value, the contact parameters as the optimization values, and a set of simulation parameter values can be obtained. Then the simulation test is carried out by using the obtained parameters. By comparing the error between the simulation test value and the physical test value of the repose angle, the rationality of the optimized parameters is verified. The Optimization module in the Design expert software calculates the selected target value and parameter range values directly. For the determination and verification of parameters, we refer to literature [19] (Fang, M.; Yu, Z.; Zhang, W.; Cao, J.; Liu, W. Friction coefficient calibration of corn stalk particle mixtures using plackett-burman design and response surface methodology. Powder Technol. 2022, 396, 731–742). We have improved the description of this section to make it easier to understand.

Point 2: The test method of regression equation is used to verify the correctness of the response surface, but since the test data are all from simulation, how does the random error generate? Analysis of variance cannot be used without random error.

Response 2: Thank you for your valuable comments and suggestions. After the simulation test of repose angle, the particle pile images under the +X and +Y views were collected, the MATLAB software was used to preprocessing, gray processing, hole filling, and binarization, the boundary information of repose angle of cotton stalks was extracted. Because the repose angles were measured from different angles, random errors were generated.

Reviewer 4 Report

Separating cotton stalk particles from the residual film after crushing is an important research topic. The use of the discrete element method is helpful to the development of related research. Therefore, the author's work is meaningful. However, the paper has the following issues that need to be revised and improved.

  1. The cotton stalk was divided into three parts (xylem, pitch, and epidermis) according to the biological structure. In building the cotton stalk model, different sizes of particles were used to represent the epidermis and internal tissues. However, the authors did not consider their differences in the selection of model parameters, and the corresponding mechanical properties were not verified. The model's validity cannot be proved by measuring the rolling friction coefficient alone, and further explanation of the basis of the model construction is needed.
  2. According to the authors, the cotton stalk particle model will be used for studies related to separating cotton stalks from the film. However, the authors did not consider the interaction between the film and the cotton stalks (e.g., the respective content of cotton stalks and the film, the winding of the film on the cotton stalks, etc.) when building the model. Further analysis of the original material properties is recommended.
  3. Appropriate simplifications can facilitate modeling and simulation analysis. However, for the object of the authors' proposed study (residual film and cotton stalk mixture), the simplifications in this paper seem to be overly simplistic (e.g., regular cuts, complete shapes, and minor variations in size), and standardized samples may have limitations when simulating crushed stalk. In particular, related studies did not consider the effect of film.
  4. The authors constructed a DEM model of the cotton rod particles using non-equal particles and used Hertz-Mindlin (No Slip) as a contact model between the particles. In this case, the cotton rod is used as a rigid body rather than a flexible body. The authors need to evaluate whether it still makes sense to use non-equal particles.
  5. In section 2.3.3, please add the test equipment specifications and the test procedure description.
  6. In section 2.2, the authors give equations for several parameters, while in 2.5.1, the values of the relevant parameters are given directly in a table. If possible, it is suggested to detail the connection between the two parts.
  7. Further revision of the abstract is suggested.

Author Response

Point 1: The cotton stalk was divided into three parts (xylem, pitch, and epidermis) according to the biological structure. In building the cotton stalk model, different sizes of particles were used to represent the epidermis and internal tissues. However, the authors did not consider their differences in the selection of model parameters, and the corresponding mechanical properties were not verified. The model's validity cannot be proved by measuring the rolling friction coefficient alone, and further explanation of the basis of the model construction is needed.

Response 1: Thank you for your valuable comments and suggestions. In this paper, the repose angle was selected as the response value to calibrate the contact parameters between particles, and the test was mainly the contact mechanics between the particle epidermis. The internal structure had little effect on the simulation results, so different sizes of particles were used to represent the epidermis and internal tissues. We also researched the effect of internal particle bonding parameters on the shear properties of cotton stalks, and the results obtained will be published later.

Point 2: According to the authors, the cotton stalk particle model will be used for studies related to separating cotton stalks from the film. However, the authors did not consider the interaction between the film and the cotton stalks (e.g., the respective content of cotton stalks and the film, the winding of the film on the cotton stalks, etc.) when building the model. Further analysis of the original material properties is recommended.

Response 2: Thank you for your valuable comments and suggestions. The construction of the DEM model of the film is still under test, and multi-material coupling simulation will be carried out in the future.

Point 3: Appropriate simplifications can facilitate modeling and simulation analysis. However, for the object of the authors' proposed study (residual film and cotton stalk mixture), the simplifications in this paper seem to be overly simplistic (e.g., regular cuts, complete shapes, and minor variations in size), and standardized samples may have limitations when simulating crushed stalk. In particular, related studies did not consider the effect of film.

Response 3: Thank you for your valuable comments and suggestions. The paper only constructed the DEM model of cotton stalk and calibrated the contact parameters between cotton stalks. In the actual test, the chopped cotton stalks have irregular shapes and incisions, and the DEM model that is completely consistent with the actual cotton stalks cannot be constructed in the software, so the model is simplified. The construction of DEM model of film is still in the test. In the future, the coupling simulation of the film and cotton stalk mixtures will be carried out, and the parameter setting will be optimized according to the law of actual test.

Point 4: The authors constructed a DEM model of the cotton rod particles using non-equal particles and used Hertz-Mindlin (No Slip) as a contact model between the particles. In this case, the cotton rod is used as a rigid body rather than a flexible body. The authors need to evaluate whether it still makes sense to use non-equal particles.

Response 4: Thank you for your valuable comments and suggestions. Because the structural characteristics and material properties of the film and the cotton stalk are quite different, the stiffness of the cotton stalk is larger than that of the film. In order to improve the simulation efficiency, the cotton stalk model is simplified and defined as rigid body.

Point 5: In section 2.3.3, please add the test equipment specifications and the test procedure description.

Response 5: Thank you for your valuable comments and suggestions. The author researched the coefficient of restitution of chopped cotton stalk in the early stage, and the related results were published in literature [43] ( Zhang, B.; Liang, R.; Li, J.; Li, Y.; Meng, H.; Kan, Z. Test and Analysis on Friction Characteristics of Major Cotton Stalk Cultivars in Xinjiang. Agriculture 2022, 12, 906.) Due to the complexity of the test procedure, in order to make the article more concise, the measurement of the oefficient of restitution is introduced in the form of references.

Point 6: In section 2.2, the authors give equations for several parameters, while in 2.5.1, the values of the relevant parameters are given directly in a table. If possible, it is suggested to detail the connection between the two parts.

Response 6: Thank you for your valuable comments and suggestions. In Section 2.2, the measurement methods of different parameters are introduced. We measured the elastic modulus, density, Poisson 's ratio of cotton stalk, the coefficient of restitution, static friction coefficient and rolling friction coefficient between cotton stalk and steel through physical tests. The parameters in the table in Section 2.5.1 are obtained by the measurement method in Section 2.2 ( except intrinsic parameters of the steel )

Point 7: Further revision of the abstract is suggested.

Response 7: Thank you for your valuable comments and suggestions. According to your suggestion, the abstract has been revised in the paper.

Round 2

Reviewer 4 Report

In response to the last comments, the authors have made some responses and changes, but some issues still need to be substantially addressed. There are some critical issues with the model, and the related research is still in progress. I understand this paper is only about cotton stalk modelling and does not involve film or their interactions. The authors should optimize the content, such as the INTRODUCTION, to explain clearly why this study was conducted. Make the research logic of the article more clear, scientific and reasonable.

Author Response

Thank you for your comments. We have made the following modifications according to your suggestions. In addition, I have made other modifications to the manuscript, so we sincerely  hope that you can review this manuscript again and make valuable suggestions. We believethis manuscript could provide readers with more valuable information. Thank you!

Point 1: In response to the last comments, the authors have made some responses and changes, but some issues still need to be substantially addressed. There are some critical issues with the model, and the related research is still in progress. I understand this paper is only about cotton stalk modelling and does not involve film or their interactions. The authors should optimize the content, such as the INTRODUCTION, to explain clearly why this study was conducted. Make the research logic of the article more clear, scientific and reasonable.

Response 1: In the third paragraph, we summarized the extensive research conducted by researchers on DEM model construction, simulation parameters and bonding parameters calibration for various types of crop stalks. In addition, to the best knowledge of the authors, there is no publication of research on the research of the calibration for cotton stalks. At the same time, separation from residual film through conventional test methods makes it difficult to accurately analyze its force and movement. Therefore, this paper carried out the construction of discrete element model and parameter calibration of cotton stalks, it is hoped that this study can provide reference for the research of membrane separation technology and the analysis of the interactions between cotton stalks and residual film or separation machinery. According to your suggestion, the INTRODUCTION has been modified to explain why this study was conducted. See line 59-63.